# Dietary Supplements for Female Infertility: A Critical Review of Their Composition

**DOI:** 10.3390/nu13103552

**Published:** 2021-10-11

**Authors:** Amerigo Vitagliano, Gabriel Cosmin Petre, Francesco Francini-Pesenti, Luca De Toni, Andrea Di Nisio, Giuseppe Grande, Carlo Foresta, Andrea Garolla

**Affiliations:** 1Department of Women and Children’s Health, University of Padua, 35122 Padua, Italy; amerigo.vitagliano@gmail.com; 2Unit of Obstetrics and Gynecology, Madonna della Navicella Hospital, Chioggia, 30015 Venice, Italy; 3Unit of Andrology and Reproductive Medicine & Centre for Male Gamete Cryopreservation, Department of Medicine, University of Padova, 35128 Padova, Italy; gabriel.petre@rocketmail.com (G.C.P.); detoni.luca@gmail.com (L.D.T.); andrea.dinisio@gmail.com (A.D.N.); grandegius@gmail.com (G.G.); carlo.foresta@unipd.it (C.F.); 4Department of Medicine, Clinical Nutrition Unit University of Padova, 35128 Padova, Italy; francescofrancini@yahoo.it

**Keywords:** fertility, dietary supplements, female reproduction, oocyte quality, pregnancy rates, spontaneous ovulation, assisted reproduction techniques

## Abstract

Infertility is the condition of about 15% of couples that cannot get a conception after one year of unprotected sexual intercourse. In females, the reduced reproductive capacity underlies the most varied causes. Dietary supplements (DS) might be used to improve the pregnancy rate and a wide range of DS are proposed today to support female fertility. Although many authors demonstrated the positive effect of some of these products, the real efficacy of this approach is still debated. In order to evaluate the potential efficacy of DS for female infertility, we analysed the products marketed in Italy, using an original approach. A review of literature was performed to evaluate the effect of nutraceuticals on various female reproductive outcomes and to detect the minimal effective daily dose (mED) able to improve at least one of these. Thereafter, we conceived a formula to classify the expected efficacy of each DS. Each DS was scored and included into three classes of expected efficacy: higher, lower, and none. Ten out of 24 supplements (41.7%) resulted in the higher and 8 (34.3%) in the lower efficacy group, the remaining 6 DS (25.0%) were expected to have no efficacy. DS marketed in Italy are usually blends of many substances that are frequently employed at a negligible dose or without any evidence of efficacy. These findings raise serious doubt about the potential effectiveness of most commercial DS for female infertility.

## 1. Introduction

Infertility is a global health problem affecting 20–30% of the female population of reproductive age in modern society [1,2]. The World Health Organization (WHO) defines infertility as a medical condition characterized by the failure to conceive after 12 months of unprotected sexual intercourse [3].

A plethora of systemic and gynecological conditions can affect the female reproductive system and potentially lead to infertility, including polycystic ovarian syndrome (PCOS), endometriosis, premature ovarian failure and pelvic inflammatory disease [4,5]. Besides organic diseases, lifestyle factors such as an unbalanced nutrition and unhealthy diet may interfere with the physiological reproductive functions in women [5,6].

The most common cause of female infertility is ovulatory dysfunction. Ovulatory disorders account for approximately 25% of infertility diagnoses; 70% of women with anovulation have PCOS [7]. Such a syndrome is a multifactorial disorder and is characterized by a combination of clinical (anovulation and hyperandrogenism), biochemical (excessive androgen and luteinizing hormone concentrations), and ovarian morphological (polycystic ovaries) features [8]. Even though PCOS etiology is complex and controversial, including genetic, environmental, and lifestyle factors, the role of insulin resistance is a key etiological component. According to these premises, several nutrients have been proposed as new therapeutic strategies for PCOS infertile patients [9].

Among the infertility causes, endometriosis is a chronic gynecological inflammatory disease characterized by the presence of functional endometrial glands and stroma outside of the uterine cavity. It affects 7–10% of women of reproductive age and up to 50% of women with infertility [10]. Although the pathophysiological mechanisms of endometriosis have not been completely explained, several pieces of data demonstrated that endometriosis is characterized by a systemic inflammatory pattern and the role of impaired natural immune system in endometriosis [11]. Moreover, endometriosis pathogenesis involves environmental factors, including dietary habits and nutrition components [12]. Since the development of endometriosis requires alterations in several biological pathways, several nutrients have been proposed as supplementary treatment for patients with endometriosis-associated infertility [12].

According to the Centers for Disease Control and Prevention (CDC) Report on Assisted Reproductive Techniques (ART), unexplained infertility arises for 11% of infertile couples requiring ART [13]. Although it is universally accepted that nutrition and life-style factors, such as diet, exercise and obesity, affect reproductive performance, preconceptional nutritional care is often inadequate. Recent studies have demonstrated that women affected by unexplained infertility may have undiagnosed dietary imbalances which negatively affect fertility [14].

In order to improve their chances of pregnancy, many women request a treatment or self-medicate with a variety of adjuvant therapies, including dietary supplements (DS) with variable ingredients [15]. In a recent study on women who were about to embark into in vitro fertilization (IVF) in a fertility center in the United Kingdom, 55% of them reported the use of some kind of DS with various ingredients. In addition, 16% of patients declared taking folic acid only [16]. The overall value of the European supplement market in 2019 reached 13.2 billion euros, and Italy is the main market with a value share of 27% (3.56 billion euros in 2019, with an increase of 4.3% compared to 2018). In clinical practice, 28.6 million supplements were prescribed in 2019, and the gynecological area reached 14% of prescriptions [17]

The rationale of using DS is to supply the deficiency of those substances or to reach the optimal dose (i.e., minerals, vitamins, carbohydrates, fatty acids and proteins) which may exert a potential positive effect on different fertility targets, such as hormonal balance, ovulation, oocyte quality, embryo quality and hopefully on the likelihood of achieving pregnancy [18,19,20]. In recent years, a growing use of DS for female infertility has been recorded, leading to the expansion of a multi-million euros industry manufacturing DS with different ingredients at different doses [21,22]. DS formulations, differently from pharmaceuticals, are exempt from patent protection and governmental sanction [23]. Notably, there are still no guidelines about the use of DS for improving female fertility, generating considerable uncertainties among prescribers. Presently, the only recommended supplement for women seeking pregnancy is folic acid, with the primary aim to prevent fetal neural tube defects [24].

In a recent critical review on DS for male infertility [25], we found that a non-negligible number of products on the market included substances without proven benefits or below the minimal effective dose. A critical review on DS for the female counterpart is still lacking. Over that background, here we present the first critical analysis on the composition of DS for female infertility, using the Italian market as a sample. Therefore, the aim of this study was to critically evaluate the potential efficacy of each DS, in particular, the effect of various commercial formulations on the female reproductive outcomes.

## 2. Materials and Methods

To identify all the DS currently marketed in Italy for female infertility, we referred to the register available on the website of the Italian Ministry of Health [26], updated on 1 June 2021. A list of all DS was obtained, and all products were included in an Excel folder. For each DS, information about ingredients (with dosage) was reported.

Subsequently, we performed a systematic literature review of studies evaluating the effectiveness of all the identified ingredients on the reproductive outcomes of infertile women (i.e., clinical pregnancy rate, ongoing pregnancy rate, live birth rate) and on surrogate markers of fertility (oocyte quality, embryo quality, fertilization rate). The literature search was conducted in Google Scholar, MEDLINE, Scopus, EMBASE, and Cochrane Library registers until 28 June 2021 following the Preferred Reporting Items for Systematic Reviews and Meta-Analysis statement (PRISMA). Only randomized clinical trials (RCTs) were included. In order to minimize detection bias, only studies evaluating the effectiveness of single substances alone or in combination with a maximum of three ingredients were considered. The key terms used for the search were: female infertility OR female reproduction OR female infertility AND supplements or ingredients. Figure 1 displays the flow diagram used for the selection of eligible studies. Selected articles allowed the identification of effective ingredients (EI), namely those substances considered as beneficial for female infertility based on available evidence. To establish the potential efficacy of each EI, we considered only those having at least one RCT demonstrating a positive, negative or no effect on any of the above-mentioned outcomes. Significance was set at a *p*-value <0.05. Regarding the daily dose of each EI with nutrient characteristics, we referred to the tolerable Upper intake Level (UL) as reported in the Dietary Reference Intake (DRI).

We considered the lowest effective dose reported in RCTs for each EI as the minimal effective daily dose (mED) able to improve any female reproductive outcomes. In a recent article, considering both RCTs and reviews on DS for male infertility, we suggested a formula derived from studies of Kuchakulla et al. [27], in order to evaluate the possible efficacy of DS based on their composition.

Briefly, the supplement ingredients were evaluated using an adapted version of the scoring system developed by The American Heart Association for the assessment of scientific evidence from clinical trials by Budoff et al. 2006 [28]. Our scoring system assigned a grade (A, B, C or D) to each ingredient based on the level of published evidence. Ingredients were qualified as evidence level A if they showed a positive impact on female infertility in multiple randomized control trials (RCTs). Level B was assigned if a positive impact was demonstrated by a single RCT. Level C was assigned if multiple RCTs showed conflicting results for the same outcome, and level D was designated for ingredients showing negative effects or a lack of evidence. Subsequently, a score was assigned to each ingredient based on its level of evidence (*A* = 5, *B* = 3, *C* = 1, *D* = −1).

Those scores were added together for each dietary supplement based on its ingredients. Then, this score was weighted for the total number of ingredients included (*N*). In order to correct this value for the number of ingredients of categories *A* and *B*, the above score was multiplied for the sum of class *A* ingredients plus half (as a proxy of their lower efficacy) the number of class *B* ingredients (*A* + *B*/2) plus one (to avoid zeros): (1)Score=5A+3B+C−DN×1+A+B2

Finally, we classified DS into three categories of expected efficacy: high expected efficacy (corrected score ≥ 4), low expected efficacy (≥1 corrected score < 4) and no expected efficacy (corrected score < 1).

**Figure 1 nutrients-13-03552-f001:**
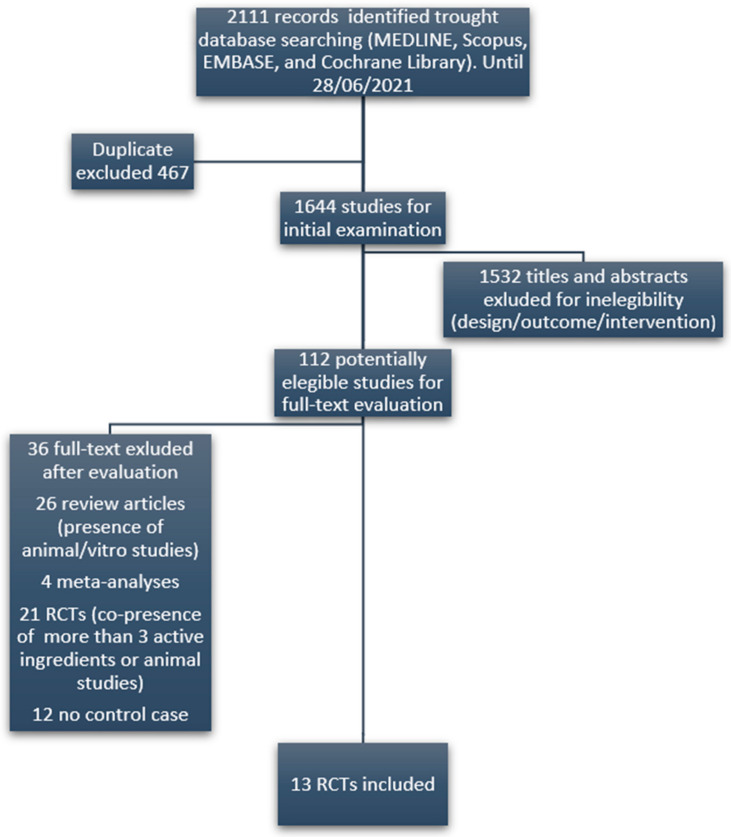
Flow diagram of the selection of eligible papers.

## 3. Results

We identified a total number of 24 DS marketed in Italy for female infertility, including a combination of 38 different ingredients. The literature review yielded 112 potentially eligible studies evaluating the efficacy of those ingredients. 99 studies (grouped into five categories) were excluded (Figure 1), resulting in 13 studies (RCTs) finally included in the review. From the results of the literature review, it emerged that 30 out of 38 ingredients had no reported efficacy for female infertility. The complete list of the only eight ingredients with a recognized clinical evidence of efficacy, the respective references, the associated clinical outcome, mED, and employed daily doses, are summarized in Table 1. In detail, five EI, respectively melatonin, folic acid, myo-inositol, *N*-acetyl cysteine (NAC) and carnitine, had evidence of efficacy supported by at least two RCTs. For the remaining ingredients, such as D-chiro inositol, CQ10 and Vitamin D, only one reference was found. Interestingly, in clinical trials using melatonin as a fertility enhancer, the dose administered was higher than that admitted by the European Food Safety Authority (EFSA).

The characteristics of the 24 DS marketed in Italy with the aim to support female fertility are summarized in Appendix A, which reports the respective composition, recommended daily dose, grade of evidence for each ingredient, and score of efficacy are reported for each DS. All the supplements contained mixtures of EI ranging from 2 up to 14 substances, with an average number of more than five ingredients. Eighteen out of 24 supplements (75.0%) contained at least one ingredient without any evidence of efficacy on female reproductive outcome. In 21 formulations (55.3%), there were ingredients with a dose below the mED. In particular, one supplement (DS 16) counted 14 ingredients, of which 10 lacking a proven efficacy, three dosed below the mED, and only one present at a dose associated with an expected efficacy. Only one DS (number 4 in Appendix A) contained exclusively ingredients with demonstrated efficacy and satisfying mED.

Among DS, the most used ingredient was the inositol (present as di-chiro and/or myo-inositol isomers), followed by folates, Vitamin D3, tocopherols, vitamin C and vitamin B12. These five molecules were used in more than the 60% of formulations, whereas each of all the remaining ingredients was found in less than 10% of products.

Based on our scoring approach, ten out of 24 supplements (41.7%) were included in the group of higher expected efficacy, eight (34.3%) in the lower efficacy group and the remaining six supplements (25.0%) in the group with no expected efficacy (Figure 2).

## 4. Discussion

To the best of our knowledge, this study represents the first critical analysis of DS formulation used for female infertility. Likely to what was noticed regarding DS for male infertility, here we found that DS for female infertility mainly contains ingredients with poor evidence of efficacy. Notably, about 80% of DS contained one or more ingredients with no evidence of effects. Moreover, many ingredients with expected efficacy (i.e., evidence of effect from clinical trials) were dosed under their mED in specific DS, raising perplexities about the effects of those DS on female fertility.

Among the DS evaluated in our study, the most common ingredient was myo-inositol (contained in 87.5% of DS). Inositols comprise a family of sugar alcohols that embraces nine stereoisomers, of which myo-inositol and D-chiro inositol are the most common isoforms in eukaryotic cells [42,43]. These molecules take part in a variety of functions and signaling pathways including reproduction, cell growth, and survival [44,45]. In relation to female fertility, it has been demonstrated that inositols facilitate the ovulatory processes through insulin-sensitizing action [46,47]. Moreover, myo-inositol was found to exert a positive effect on the in vitro maturation of oocytes in rats, suggesting a putative positive effect on oocyte quality and embryo development [48]. Additionally, a recent review of RCTs [49] showed that the daily use of 4 g myo-inositol was effective in reducing the gonadotropin dose and stimulation window in women undergoing IVF. These data subtend a specific action as a gonadotropin-sensitizer agent at the ovary level. However, our study found that myo-inositol levels reached the mED (4 g/day) only in 13.4% of DS. In the remaining supplements containing myo-inositol, its concentration was 2 g or lower. D-chiro inositol reached or exceeded the mED (300 mg) in other four products and was under-dosed in four supplements.

Folic acid was the second most common ingredient, included in 83.0% of the evaluated DS. In four supplements, its dosage did not reach the mED (400-mg dose). This aspect is relevant as WHO strongly recommends all women of reproductive age to a daily use of 400 mg of folic acid, in addition to consuming food with folate from a varied diet, in order to prevent neural tube defects (NTDs) [50]. Nonetheless, periconceptional use of folic acid was shown to reduce the incidence of non-NTD birth defects including cleft palates, upper limb reduction deficits, and genitourinary defects [51]. Thus, those DS containing low doses of folic acid (<400 mcg) may be non-protective both for NTDs and non-NTD birth defects.

Vitamin D3 was included in about 30% of supplements. Vitamin D is a fat-soluble steroid hormone with several physiological autocrine, paracrine, and endocrine functions in the female reproductive system [52]. This hormone modulates follicle recruitment through the regulation of anti-Mullerian hormone secretion [53,54]. Additionally, this substance is involved in the regulation of ovarian and endometrial cell proliferation [52,55]. However, the women’s serum levels of vitamin D do not seem to influence IVF outcome [56], and the effects of vitamin D3 supplementation on female fertility are still unknown. In this study, we observed that all the analyzed supplements were underdosed with respect to Vitamin D3, raising some concerns about the expected efficacy of these ingredients in commercial formulations.

Melatonin was contained in two DS. Melatonin is a low-molecular weighted indoleamine with multiple biological functions in humans, including regulation of biological rhythms, reproduction, immune and metabolic functions [57,58]. With respect to female fertility, melatonin supplementation was found to exert a positive effect on oocyte quality, embryo quality and luteal function [59,60]. For these reasons, it is currently considered as a promising EI for female infertility [61]. Nevertheless, our critical analysis revealed that, when melatonin was included in supplements, its concentration was three times lower than mED (1 mg vs. 3 mg). This choice depends on the fact that the Italian Ministry of Health in 2013 lowered the admitted dose of melatonin in DS from 5 mg/day to 1 mg/day, according to EFSA. This EFSA position is based on the evidence that melatonin is effective on sleep disorders at a dose below 1 mg [62]. Currently, no study on reproductive function using this low dose is available.

NAC was included in a single nutraceutical product. It has antioxidant, mucolytic and insulin-sensitizing properties [63,64]. For these reasons, NAC is commonly used as an adjuvant for the treatment of different conditions, including lung, heart and hepatic diseases [65,66]. In the field of gynecology, NAC supplementation has been demonstrated to enhance spontaneous ovulation in women with polycystic ovarian syndrome, as well as to improve oocyte and embryo quality in infertile women undergoing IVF [67,68,69].

CoQ10 is a free radical scavenger acting as an antioxidant agent in the mitochondrial respiratory chain and playing a crucial role in the regulation of energy metabolism [70,71]. Additionally, CoQ10 plays a role in the protection of female gametes from oxidative stress, as demonstrated by its age-declining concentration in the follicular fluid [72]. Its supplementation was associated with an improvement of oocyte quality in both animal and human studies, suggesting a potential role for this molecule as adjuvant in the prevention of post-ovulatory aging in infertile women [73]. Both NAC and CoQ10 were dosed below mED.

Regarding many other nutrients included in evaluated DS, we could not find any scientific evidence supporting their use in women seeking pregnancy; therefore, we can consider with expected efficacy only the eight EI included in Table 1.

The main points of strength of our study were the original design and rigorous methodology. The choice to include only evidence from RCTs and meta-analyses was made in order to minimize bias in the critical evaluation of DS. The main limitation of this study is the restricted focus of research on DS based on the Italian market.

In conclusion, we demonstrated that the majority of DS marketed in Italy for female infertility had at least one or more ingredients with no proven effect. In addition, ingredients with proven efficacy were frequently underdosed and/or combined with a variety of nutrients with unknown effects on female fertility. These findings raise serious doubts about the potential effectiveness of most commercial DS for female infertility. Our approach could be applied in the evaluation of DS marketed in other fields and in other countries.

## Figures and Tables

**Figure 2 nutrients-13-03552-f002:**
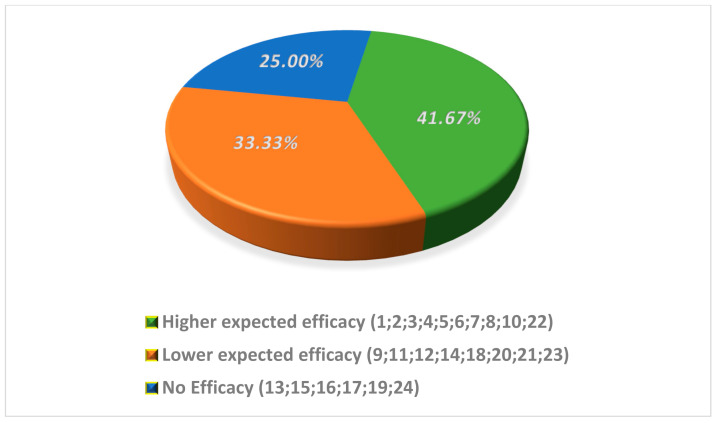
Distribution of supplements in classes of expected efficacy.

**Table 1 nutrients-13-03552-t001:** Active ingredients with evidence of efficacy, references, evaluated fertility outcomes, employed daily doses and minimal fertility effective doses (mED). **AMH**: anti-Müllerian hormone; **NAC**: *N*-Acetil-Cysteine; **CQ10**: Coenzyme Q 10; A and B refer to the level of scientific evidence for each individual active ingredient. Each molecule can receive the highest or lowest level whether the employed dosage reaches the mED or not.

Active Ingredients	References	Outcome Evaluated	Employed Daily Dose	Minimal Effective Dose (mED)
Melatonin—A/B	[29] Pacchiarotti et al.	Oocyte and embryo quality	3 mg	3 mg
	[30] Espino et al.	Ox status, oocyte quality	3 mg	
	[30] Espino et al.	Ox status, oocyte quality	6 mg	
Folic acid—A/B	[31] Ciotta et al.	Oocyte quality	400 mcg	400 mcg
	[32] Papaleo et al.	Oocyte quality	400 mcg	
	[29] Pacchiarotti et al.	Oocyte and embryo quality	400 mcg	
Myo-inositol—A/B	[31] Ciotta et al.	Oocyte quality	4 g	4 g
	[33] Unfer et al.	Oocyte and Embryo quality	4 g	
	[32] Papaleo et al.	Oocyte quality	4 g	
	[29] Pacchiarotti et al.	Oocyte and embryo quality	4 g	
D-chiro-inositol—B/C	[34] Mendoza et al.	Pregnancy rate	300 mg	300 mg
NAC—A/B	[35] Badawy et al.	Ovulation rate	1.2 g	1.2 g
	[36] Nasr et al.	Pregnancy rate	1.2 g	
	[37] Cheraghi et al.	Oocyte and Embryo quality	1.8 g	
CQ10—B/C	[38] Xu et al.	Fertilization rate and embryo quality	600 mg	600 mg
Carnitine—A/B	[39] Kitano et al.	Embryo quality	1 g	1 g
	[40] Ismail et al.	Pregnancy rate	3 g	
Vitamin D—B/C	[41] Dastorani et al.	AMH level	89 mcg	89 mcg

## Data Availability

Publicly available datasets were analyzed in this study. This data can be found in Appendix A.

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
