# Peer review of "Dietary Supplements for Female Infertility: A Critical Review of Their Composition"

_nutrients, 2021, doi:10.3390/nu13103552_

Round 1

Reviewer 1 Report

This is the revised article following my previous report

Reviewer 2 Report

The submitted manuscript “Dietary supplements for female infertility: a critical review of their composition was read with care and please find the comments below.

The manuscript is well-written, easy to read & understand & comprehend.

There will be minor concerns to make the manuscript get into better shape.

12 RCTs and 38 ingredients, 24 supplements included in the manuscript. The number of the papers/reviews/meta-analysis seems to be improved to make “a critical review”.

The quality of Figure 1 and Figure 2 definitely needs improvement, the existing format is not acceptable. Also, Table 1 has no legend, and the figures’ legend is not explanatory especially the numbers in Figure 2 are confusing. You can also see formatting error in the pie diagram no efficacy group percentage is not seen due to the formatting.

Overall, structure-wise and language-wise, there are no concerns.  

This manuscript is a resubmission of an earlier submission. The following is a list of the peer review reports and author responses from that submission.

Round 1

Reviewer 1 Report

The measurement of DS efficacy needs improving. There is no mention to pregnancy or live birth rates. These are DS recommended for fertility. the only efficacy measured should be live birth or at least pregnancy rate. In these paper some Ds are deemed efficient without improving the chance of conception.

Clear outcomes need to be drawn and discussion/results to be re-written based on the above.

Author Response

Thank you for this relevant comment, which calls in question the value of surrogate endpoints in reproductive medicine. The aim of our study was to build an evidence-based paradigm to a conscious application of nutraceuticals basing on a systematic literature review. However, we must specify that any systematic review, in turn, relies on the results of primary studies. In this respect, we found that the majority of primary studies used surrogate measures of treatment response to nutraceuticals. The benefits of choosing to study surrogate variables instead of direct outcomes in clinical trials (egs. Clinical pregnancy rate, live birth rate) are that they may shorten the period of study, lower the sample size required and lower the costs of the study. Ideally, if there is a strong correlation between change in the surrogate variable (i.e. Ovulation in those women suffering from anovulation) and the primary clinical end-point (i.e. pregnancy rates), the effectiveness of therapy on the primary end-point can be quantified by estimating the proportion of the treatment effect explained along with its standard error (Lin DY, Fleming TR and De Gruttola V,1997, Estimating the proportion of treatment effect explained by a surrogate marker. Stat Med 16,1515–1527).

Although we agree that direct endpoints are the ideal measures of effect of a certain treatment, we hope the revewer will appreciate our efforts to evaluate the available data on this topic, as well as we hope he will recognise that surrogate endpoints may be valuable when direct measures of effect of a certain treatment are lacking.

Reviewer 2 Report

The authors carried out a literature review combined with dietary supplements market investigation to determine the relationship between the potential efficacy of dietary supplements for female infertility in Italy and scientific proofs of its action. This topic is of great importance to know if the evidence for incorporating each supplements ingredient is convincing, probable or limited. Nevertheless, the importance of this research is not entirely clear and elucidated in the manuscript.

Below authors can find several comments and suggestions to improve the manuscript:

The word approach appears three times in three consecutive sentences (lines 14-18) in the abstract.

The introduction should provide sufficient background on the topic. In my opinion the introduction section is too basic and insufficiently shows the scale of the problem and the causes of infertility.

I would expect more detailed information on the etiology of infertility.

The scale of taking dietary supplements by infertile women should also be discussed based on the Italian population (the market of dietary supplements in Italy was investigated).

When the epidemiology of female infertility is commented current references should be added.

Figure 1 - flow diagram of the study should be modified to horizontal orientation. Now it takes too much place. Whats mean 103 just below the last windows with the text "13 RTCs included"

The Table 1 caption is: " Active ingredients with evidence of efficacy, references, evaluated sperm parameters, em-143 ployed daily doses and minimal fertility effective doses". The sperm parameters describe male, not female, fertility. Additionally, in the Table 1 caption, the abbreviation A/B and B/C should be explained.

Line 145, the space between "Q10" and "The characteristic..." should be added. I assume that probably "The characteristic..." in line 145 should be the beginning of the new line/paragraph?

In my opinion, almost 60% of the manuscript are nearly empty tables. I suggest considering transferring tables with the composition of supplements to supplementary materials and only describe them in the text in more detail.  I also suggest adding the tables with research papers results describing the effectiveness of each ingredient used in the dietary supplements. 

Discussion should be significantly improved and divided into nutritional factors influencing female fertility with proven effectiveness and inconclusive evidence of effectiveness. Currently, the authors did not refer to factors other than active ingredients with evidence of efficacy in the discussion. The authors should try to explain the rationale to use each ingredient. What action of the individual components are mentioned by the manufacturers in the leaflet?

Authors should also pay more attention to explain the physiological role of described dietary supplements ingredient based on in vitro, animal model studies and human involved clinical studies.

The authors did not refer to the possible side effects of taking supplements, the effectiveness of which is unconfirmed.

Author Response

Thank you for the constructive comments that have helped us improve our work.

The first half of the considerations were carried out throughout the manuscript in accordance with the suggestions obtained by the reviewer.

Figure 1 - flow chart we thought that leaving it vertically is much more usable as it naturally gives the idea of ​​how many studies we have analyzed to arrive at the RCTs considered eligible

Table 1 and the error on line 145 have been corrected.

Table 2 has now been removed from the main text and inserted as supplementary material

We have better elucidated both within the results and in the discussion that the only active ingredients that can be considered useful, given that they have supporting literature, are the 8 that we have included in table 1

Regarding the last three suggestions of the reviewer: 1) we have not treated every single active ingredient from the point of view of physiology as the aim of our study is to evaluate the commercial formulations as a whole and not to make a systematic review of every single molecule.

2) We have clearly specified that for our purpose (as suggested by the American Heart Associasion, creators of the equation from which we started) we have not considered other studies other than RTCs, which is why we have not included / treated other types of studies.

3) Using food supplements and following the doses recommended by the manufacturers can probably exclude side effects as it is not possible to introduce pharmacological dosages of nutrients in a food supplement.